# Metabolic analysis of amino acids and vitamin B6 pathways in lymphoma survivors with cancer related chronic fatigue

Alexander Fosså[1,2,3]*, Knut Halvor Smeland[1,2], Øystein Fluge[4], Karl Johan Tronstad[5], Jon Håvard Loge[6], Øivind Midttun[7], Per Magne Ueland[7,8], Cecilie Essholt Kiserud[1]

1 Department of Oncology, National Advisory Unit on Late Effects after Cancer Treatment, Oslo University Hospital, Oslo, Norway, 2 Department of Oncology, Oslo University Hospital, Oslo, Norway, 3 KG Jebsen Center for B-cell malignancies, Oslo University, Oslo, Norway, 4 Department of Oncology and Medical Physics, Haukeland University Hospital, Bergen, Norway, 5 Department of Biomedicine, University of Bergen, Bergen, Norway, 6 Regional Centre for Excellence in Palliative Care, Oslo University Hospital, Oslo, Norway, 7 Bevital AS, Bergen, Norway, 8 Department of Clinical Science, University of Bergen, Bergen, Norway

* aff@ous-hf.no

**Data Availability Statement:** The data which have been included with the paper and its Supporting Information is the full extent of the data allowed to be shared in accordance with Norwegian

## Abstract

Chronic cancer-related fatigue (CF) is a common and distressing condition in a subset of cancer survivors and common also after successful treatment of malignant lymphoma. The etiology and pathogenesis of CF is unknown, and lack of biomarkers hampers development of diagnostic tests and successful therapy. Recent studies on the changes of amino acid levels and other metabolites in patients with chronic fatigue syndrome/myalgic encephalopathy (CFS/ME) have pointed to possible central defects in energy metabolism. Here we report a comprehensive analysis of serum concentrations of amino acids, including metabolites of tryptophan, the kynurenine pathway and vitamin B6 in a well characterized national Norwegian cohort of lymphoma survivors after high-dose therapy and autologous stem cell transplantation. Among the 20 standard amino acids in humans, only tryptophan levels were significantly lower in both males and females with CF compared to non-fatigued survivors, a strikingly different pattern than seen in CFS/ME. Markers of tryptophan degradation by the kynurenine pathway (kynurenine/tryptophan ratio) and activation of vitamin B6 catabolism (pyridoxic acid/(pyridoxal + pyridoxal 5'-phosphate), PAr index) differed in survivors with or without CF and correlated with known markers of immune activation and inflammation, such as neopterin, C-reactive protein and Interleukin-6. Among personal traits and clinical findings assessed simultaneously in participating survivors, higher neuroticism score, obesity and higher PAr index were significantly associated with increased risk of CF. Collectively, these data point to low grade immune activation and inflammation as a basis for CF in lymphoma survivors.

regulations. The public sharing of the remainder of the data underlying this study are restricted in accordance with Norwegian law by South East Regional Committee for Medical and Health Research Ethics as the data contain potentially identifying patient information. These data are available on request to Data Protection Officer at Oslo University Hospital (personvern@ous-hf.no). In order to obtain the minimal data set for this study, please provide the approval number by the South East Regional Committee for Medical and Health Research Ethics (approval number 2011/ 1353 B).

**Funding:** This study was supported in part by research funding from the Norwegian Cancer Society and Radiumhospitalets legater. ØM and PMU are both at least partially employed by Bevital, a non-profit organization. Bevital has only paid parts of the salaries of these two coauthors, and not had any additional role in the study design, data collection and analysis, decision to publish, or preparation of the manuscript. The specific roles of these authors are articulated in the 'author contributions' section. The funder provided support in the form of salaries for authors [ØM and PMU], but did not have any additional role in the study design, data collection and analysis, decision to publish, or preparation of the manuscript.

**Competing interests:** PMU is a member of the steering board of the nonprofit foundation which owns Bevital, and R&D director of Bevital, the company that carried out biochemical analyses. The authors declare no competing financial interests. This does not alter our adherence to PLOS ONE policies on sharing data and materials.

## Introduction

Persistent fatigue is a subjective experience of tiredness, exhaustion and lack of energy that has a negative impact on daily life and functioning. It is a common symptom in a wide variety of disorders, for instance, in patients with inflammatory or infectious diseases, depression disorder and cancer [1–3]. Together with post-exertional malaise, a marked aggravation of symptoms after exercise, fatigue is also the hallmark symptom in patients with chronic fatigue syndrome/myalgic encephalopathy (CFS/ME) [4].

Whereas acute fatigue is a healthy, adaptive response to physical or mental exertion and typically resolves after rest or sleep, persistent fatigue is often disproportional to exerted activities and is generally not completely alleviated after a period of rest. The pathophysiological changes leading to persistent fatigue in different diseases are poorly understood and no specific treatment is available to ameliorate fatigue in affected individuals.

Chronic cancer-related fatigue (CF, defined as pronounced fatigue for $\geq$ 6 months) is a common and distressing late effect after cancer treatment affecting patients treated for both solid cancers and hematological malignancies [1, 5, 6]. CF has been described in 25–35% of long-term survivors of breast cancer, lymphoma or testicular cancer in Norway, compared to 11% in a national representative population [7–10]. These patients are cured of their malignancies but may suffer from other late effects where fatigue may be an associated symptom. The etiology and pathophysiology of CF in cancer survivors is largely unknown, though evidence suggests it may be multifactorial, influenced by demographic, somatic and psychological factors [5, 11]. The incidence varies with the type of cancer and treatment given and seems to increase in the presence of psychological discomfort such as anxiety, pessimism, low mood or depression. Somatic comorbidities such as endocrine abnormalities, pulmonary dysfunction or cardiovascular disease may predispose patients to CF [12–14]. In the care of survivors with CF, it is important to rule out and treat such coexisting symptoms and conditions. Some studies point to an ongoing low grade inflammatory process in patients with CF after breast cancer or lymphoma [15–17].

Despite knowledge of these associated conditions, the absence of an etiological and pathophysiological understanding of CF limits the ability to diagnose and treat CF. For example; it is not known whether the mechanisms underlying fatigue in different medical conditions are similar. Recently, large scale metabolic studies in CFS/ME have revealed underlying defects related to key pathways of metabolism that discriminate patients with CFS/ME from healthy individuals, providing a possible basis for future diagnostic tests and therapeutic interventions [18–20]. We and others found in patients with CFS/ME that circulating concentrations of amino acid and other metabolites fueling energy generation in the tricarboxylic acid were altered and energy metabolism hampered [18, 19]. Such observations have allowed generation of hypotheses as to mechanisms driving fatigue and associated symptoms in CFS/ME [21].

On this background, we recently assessed the prevalence of CF in a national cohort of adult lymphoma survivors treated with high-dose therapy and autologous stem cell transplantation (HDT-ASCT) [22]. We investigated associations between CF and disease- and treatment-related characteristics, psychological factors and objectively measured somatic health, including cardiorespiratory fitness and selected cytokines. CF is a prevalent long term condition affecting about 30% of these survivors and our data suggest ongoing low grade inflammation as a pathogenic mechanism. With the aim to explore further the underlying metabolic defects driving CF in cancer survivors, we now extend these studies to metabolic analyses as previously done in CFS/ME, comparing individuals with or without CF. We report here metabolic analyses in lymphoma survivors with a median follow-up of 10 years after HDT-ASCT.

## Patients and methods

### Patients

The study was part of a national multicenter cross-sectional study performed in 2012–2014 [23]. All survivors treated with HDT-ASCT for lymphoma in Norway from 1987 to 2008, aged ≥18 years at HDT-ASCT, resident in Norway at survey, in remission and not currently undergoing systemic therapy for active malignancy, were eligible as described earlier (n = 399) [23]. The survivors were identified through treatment records and registries at each participating center. Eligible survivors were invited by mail to complete a questionnaire and attend an outpatient clinical examination, including blood sampling and exercise testing with measurement of peak oxygen consumption (VO2peak) [22–24].

The study was approved by the South East Regional Committee for Medical and Health Research Ethics (approval number 2011/1353 B). Written informed consent was given by all participants.

Blood was drawn by venous puncture after overnight fasting on the morning of the outpatient clinical examination. Plasma glucose, C-reactive protein (CRP), albumin, triglycerides, total cholesterol, low and high density lipoprotein cholesterol were measured according to routine laboratory facilities at each hospital. For serum preparation and later metabolic analyses, tubes without gel were used, allowed to coagulate for 30–60 minutes prior to centrifugation for 15 minutes at 1000 x g at 4˚C. Serum was aliquoted into sterile tubes and frozen at -80˚C. Samples were stored at -80˚C and thawed immediately prior to analysis.

Lymphoma- and treatment-related data were obtained from patients' charts. Treatment of lymphomas in Norway, including HDT-ASCT, has followed international and national guidelines [25]. In the period 1987–1995 the conditioning regimen consisted of total body irradiation (TBI) and high-dose cyclophosphamide, and from 1996 chemotherapy only (BEAM: carmustine, etoposide, cytarabine and melphalan). The survivors were grouped according to time of HDT-ASCT: 1987–1995, 1996–2002 and 2003–2008 and primary diagnosis: Hodgkin lymphoma (HL), aggressive non-Hodgkin lymphoma (NHL) (diffuse large B-cell lymphoma, T-cell lymphomas, mantle cell lymphoma, Burkitt's lymphoma and lymphoblastic lymphoma) and indolent NHL (mostly follicular lymphoma).

### Patient reported outcomes and cytokines

Neuroticism was assessed by an abbreviated version of the Eysenck Personality Questionnaire with six items, with higher score implying more neuroticism [26]. Internal consistency assessed by Cronbach's α was 0.79. A 15-item version of the Impact of event scale (IES) was used to measure post-traumatic symptoms related to HDT-ASCT, consisting of seven items on intrusion and eight on avoidance, with responses on a six-point frequency scale for each item [27]. Internal consistencies were 0.91 for intrusion and 0.91 for avoidance. Mental distress was measured by the Hospital Anxiety and Depression Scale (HADS). Each item was scored from 0 (not present) to 3 (highly present) [28]. Internal consistencies were 0.82 and 0.87 for the depression and anxiety subscales, respectively.

Chronic fatigue was assessed by the Fatigue Questionnaire which contains 11 items concerning physical (7 items) and mental (4 items) fatigue during the last month, compared with when the respondent last felt well [29]. Each item has four response alternatives scored from 0 (better) to 3 (much worse). Two additional items cover duration and extent of fatigue. Responses were dichotomized (0 = 0 and 1; 1 = 2 and 3) and used for case definition, with CF defined as a sum score of ≥4 of the dichotomized responses with duration of ≥6 months.

Internal consistency (Cronbach's α) was 0.92 for total fatigue, 0.93 for physical fatigue and 0.80 for mental fatigue, respectively.

Measurements of Interleukin(IL)-6, IL-1β, IL-1Receptor Antagonist (RA) and Tumor Necrosis Factor(TNF)α in serum have been reported for this cohort previously. As reported, IL-1β and TNFα levels were not associated with the prevalence of CF and therefore not analyzed further. IL-6 was analytically undetectable in a substantial number of patients, and therefore dichotomized as detectable versus not detectable [22].

## Analyses of amino acids, metabolites, B-vitamers and neopterin

All analyses were done by Bevital AS, Bergen, Norway (www.bevital.no). The standard amino acids (except Arginine (Arg)), ornithine, α-ketoglutaric acid and the tryptophan (Trp) metabolite kynurenine (Kyn) were analyzed using gas chromatography–tandem mass spectrometry (GC-MS/MS), with within and between day coefficient of variation (CV) of 2%–5% [30]. Arg, homoarginine (hArg), methylated asymmetric or symmetric arginines (ADMA and SDMA) were analyzed by liquid chromatography–tandem mass spectrometry (LC-MS/MS), with within and between day CVs of 3%–12% [31]. Biomarkers related to vitamin B6 status, i.e. pyridoxal 5'-phosphate (PLP), pyridoxal (PL) and 4-pyridoxic acid (PA) and other metabolites of the kynurenine pathway, i.e. kynurenic acid (KA), anthranilic acid (AA), 3-hydroxykynurenine (HK), xanthurenic acid (XA) and 3-hydroxyanthranilic acid (HAA), picolinic acid (Pic), quinolinic acid (QA), in addition to nicotinic acid (NA), nicotinamide (NAM) and N1-methylnicotinamide (mNAM) and neopterin were quantified using LC-MS/MS with within-day and between day CVs of 2–17% [32]. Because the assay includes protein precipitation by trichloroacetic acid, which oxidizes 7,8-dihydroneopterin to neopterin, and the method measures total neopterin [32, 33].

As markers of inflammation the following indices were derived from the analysed biomarkers: PAr index was calculated as the ratio of PA divided by the sum of PLP and PL (PA/(PLP+PL)). PAr efficiently discriminates subjects with high inflammatory status [34, 35]. It is only slightly influenced by vitamin B6 intake and reflects increased vitamin B6 catabolism during inflammation. The kynurenine/tryptophan (Kyn/Trp) ratio was calculated from Kyn and Trp concentrations. The Kyn/Trp ratio has been reported to be a marker of cellular immune responses [36, 37].

## Statistics

Descriptive statistics including t-tests for normally distributed data, Mann-Whitney U tests for variables with skewed distributions and Chi-square and Fischer's exact tests for categorical variables, were used as appropriate. Correlations are reported as Spearman coefficients ($r_s$). Multivariate logistic regression analysis was performed with CF as the dependent variable, including predictors with p-value <0.10 in univariate analyses. As reported previously, subscales of HADS were excluded due to high correlation with neuroticism. PAr substituted IL-6 in the model due to highly significant bivariate correlations. Due to the explorative and hypothesis generating design of the study, the significance level was set to 0.05, and all tests were two-sided. Statistical analyses were performed using International business machines Statistical Package for the Social Sciences version 23. Figures were produced using Stata SE version 15.

## Results

### Attrition analysis and patient characteristics

Of 399 eligible survivors, 311 (78%) consented and completed the questionnaire (Fig 1). Of these, 270 attended the clinical examination. Blood tests were taken as specified in the protocol

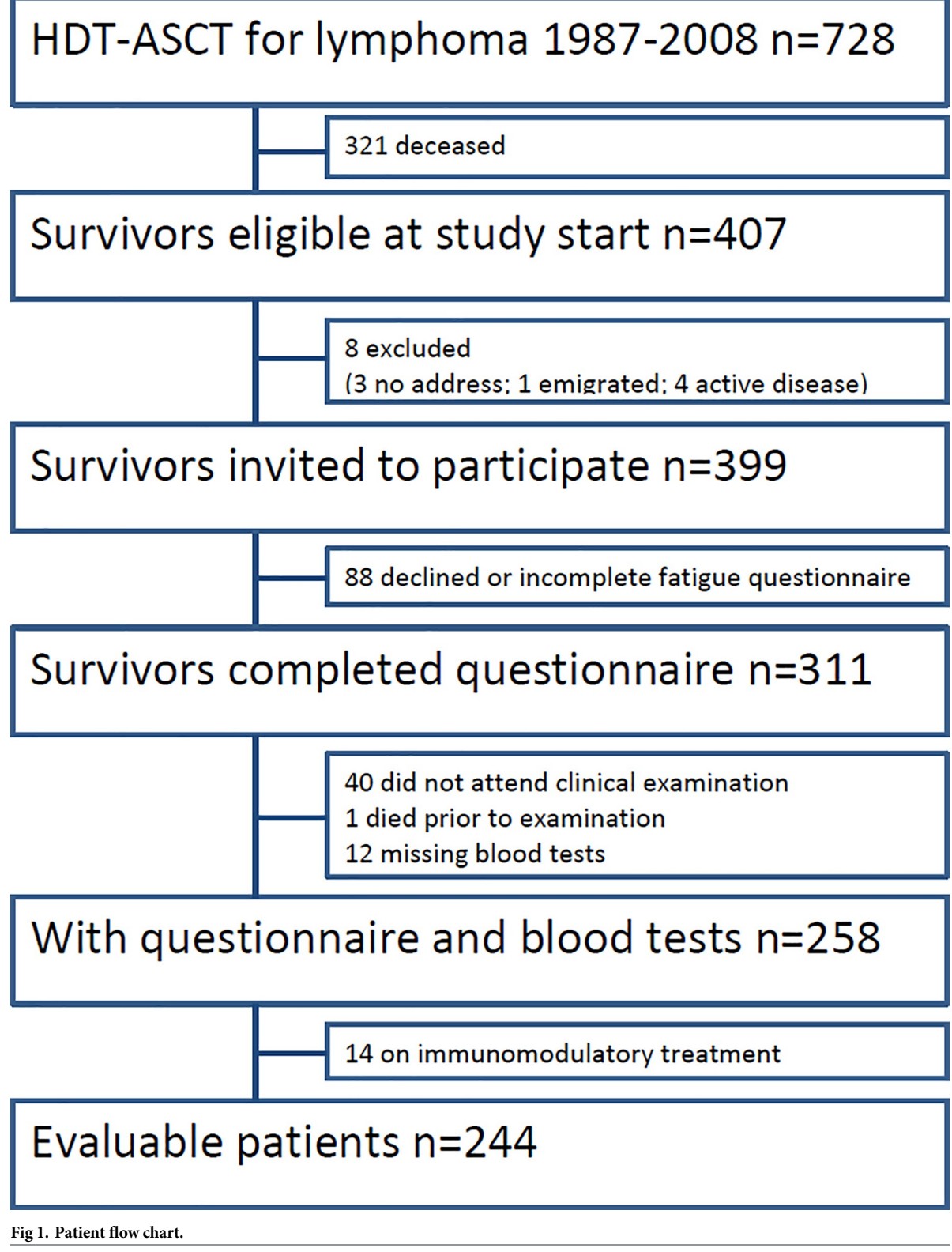

**Fig 1. Patient flow chart.**

in 258 survivors. For the final analysis, 14 patients with ongoing immune modulatory treatment were excluded (i.e. treatment with anakinra, prednisolone, cyclosporin A or intravenous immunoglobulins). The 244 (61% of the total) included participants (Table 1) were slightly older than non-participants, but there was no statistically significant difference regarding age, sex, observation time, lymphoma type or conditioning regimen. After a median follow-up of 12 years since diagnosis, the prevalence of CF was 32%. The proportions of chronically fatigued survivors by patient, disease and treatment characteristics, patient reported outcomes and findings from clinical examination are given in Table 1. There were no differences between fatigued and non-fatigued survivors concerning glucose, triglycerides or total, high or low density lipoprotein cholesterol.

## Serum amino acid concentrations

Based on findings that serum concentrations of amino acids may reveal defects in energy metabolism we analyzed serum concentrations of all standard 20 amino acids in survivors with or without CF (S1 Table). Since there were gender differences for several of the tested amino acids, the results are reported for the whole groups of survivors and for female and male patients separately. The only amino acid with a statistically significant difference in serum concentration was Trp. Survivors with CF had a mean concentration (95% confidence interval (CI)) of 73.4 (70.5–76.4) μM compared to 77.6 (75.9–79.3) μM in non-fatigued survivors (p = 0.01; Fig 2). Differences were similar in each gender, but did not reach significant p-values (S1 Table). The concentrations of all other amino acids were similar in survivors with or without CF, both when analyzed individually or when combined for groups of amino acids converted to pyruvate (Category I), to acetyl-coenzyme A (Category II) or for the anaplerotic amino acids (category III) converted to different intermediates of the tricarboxylic acid cycle.

Of the other amino acid related metabolites tested, the only significant difference was seen for the tricarboxylic acid cycle intermediate α-ketoglutaric acid (S2 Table). Survivors with CF had a mean concentration (95% CI) 9.2 (8.5–9.7) μM and non-fatigued survivors 8.4 μM (8.2–8.8; p = 0.02; Fig 2). The difference in serum α-ketoglutaric acid concentration was seen in both males and females, but not with significant p-values for sexes separately (S2 Table). There were no differences for metabolites relating to Arg metabolism and endothelial function (Arg, hArg, ADMA or SDMA) or metabolites relating to the urea cycle (ornithine and urea).

## Kynurenine pathway, vitamin B6 markers and neopterin

Since Trp is mainly degraded by the kynurenine pathway and this pathway is subject to regulation by inflammatory signals, metabolites along this degradation pathway were analyzed (S3 Table) [37]. There were again no clear differences in the concentration of the different Trp metabolites according to the presence of CF in the whole group of survivors. However, the ratio of Kyn to Trp concentrations was higher in fatigued (mean 0.029; 95% CI 0.025–0.033) than non-fatigued survivors (mean 0.026; 95% CI 0.025–0.028) with a borderline significance level of p = 0.06 (S3 Table; Fig 3). Further, there were indications of change for two of the intermediates in this pathway. The level of HK tended to be higher (p = 0.07) in the patients with CF (63.2 (0–135.5) μM) compared to those without CF (56.1 (7.9–104.3) μM). When separating the genders, a significantly higher HK level was observed among men with CF (64.9 (0–143.1) μM versus 54.5 (12.3–96.7) μM, p = 0.04), but not women. The level of HAA tended to be lower (p = 0.09) in the CF group (52.5 (11.2–93.7) μM) compared to those without CF (57.1 (19.7–94.5) μM). This effect was primarily due to a trend of difference between the female patients with and without CF (45.3 (13.4–77.2) μM versus 52.5 (14.7–90.3) μM, p = 0.07)

**Table 1. Patient characteristics according to chronic fatigue.**

| | No chronic fatigue (n = 167 | Chronic fatigue (n = 77) | P |
|---|---|---|---|
| Sex | | | 0.26 |
| Male | 109 | 44 | |
| Females | 58 | 33 | |
| Median age at diagnosis/years (range) | 42 (10–65) | 40 (17–64) | 0.66 |
| Age at survey/year (range) | 56 (25–76) | 55 (24–77) | 0.37 |
| Median time diagnosis to survey/months (range) | 152 (49–408) | 125 (43–367) | 0.54 |
| Median time diagnosis to HDT-ASCT[a]/months (range) | 15 (2–257) | 15 (2–272) | 0.99 |
| Lymphoma type | | | 0.20 |
| Hodgkin lymphoma | 31 | 21 | |
| Aggressive Non-Hodgkin lymphoma | 121 | 47 | |
| Indolent Non Hodgkin lymphoma | 15 | 9 | |
| Treatment period | | | 0.69 |
| 1987–1995 | 28 | 11 | |
| 1996–2002 | 49 | 20 | |
| 2003–2008 | 90 | 46 | |
| Ann Arbor stage at diagnosis* | | | 0.46 |
| I/II | 50 | 27 | |
| III/IV | 116 | 50 | |
| B-symptoms at diagnosis† | | | 0.48 |
| No | 110 | 47 | |
| Yes | 54 | 29 | |
| High dose regimen | | | 0.57 |
| TBI[b] + Cyclophosphamide | 27 | 10 | |
| BEAM[c] | 140 | 47 | |
| Mediastinal radiotherapy | | | 0.026 |
| No | 62 | 28 | |
| Yes | 53 | 36 | |
| Other | 52 | 13 | |
| Rituximab | | | 0.78 |
| No | 98 | 47 | |
| Yes | 69 | 30 | |
| Relapse after HDT-ASCT | | | 0.30 |
| No | 138 | 59 | |
| Yes | 29 | 18 | |
| Allogeneic SCT[d] after HDT-ASCT | | | 0.47 |
| No | 162 | 73 | |
| Yes | 5 | 4 | |
| Body mass index (kg/m$^2$) | | | 0.02 |
| <30 (not obese) | 150 | 60 | |
| ≥30 (obese) | 17 | 17 | |
| Median score HADS[e] A (range) | 3 (0–12) | 5 (0–19) | p<0.001 |
| Median score HADS D (range) | 1 (0–12) | 5 (0–15) | p<0.001 |
| Median neuroticism score (range) | 0 (0–6) | 3 (0–6) | p<0.001 |
| Median impact of event score (range) | 5 (0–60) | 13 (0–62) | p<0.001 |
| Mean CRP[f]/mg/L (SD[g]) | 5.5 (18.5) | 3.6 (6.6) | 0.22 |
| Mean plasma glucose/mmol/L (SD) | 5.9 (1.4) | 5.7 (0.9) | 0.32 |
| Mean triglycerides/mmol/L (SD) | 1.4 (0.9) | 1.3 (0.6) | 0.20 |

(*Continued*)

**Table 1.** (Continued)

| | No chronic fatigue (n = 167 | Chronic fatigue (n = 77) | P |
|---|---|---|---|
| Mean cholesterol/mmol/L (SD) | 5.3 (1.2) | 5.4 (1.2) | 0.33 |
| Mean LDL[h] cholesterol/mmol/L (SD) | 3.2 (1.1) | 3.5 (1.1) | 0.08 |
| Mean HDL[i] cholesterol/mmol/L (SD) | 1.5 (0.5) | 1.5 (0.5) | 0.69 |
| Mean albumin/g/L (SD) | 43.5 83.3) | 44.1 (2.8) | 0.14 |
| Mean VO$_2$[j] peak/L/min (SD[h]) | 2.29 (0.72) | 2.04 (0.66) | 0.02 |
| Interleukin-6 detectable‡ | 84 | 53 | 0.005 |
| Interleukin-1RA[k]/pg/mL (SD)‡ | 66.0 (96.5) | 42.0 (55.9) | 0.04 |

[a] High dose therapy with autologous stem cell transplantation

[b] Total body irradiation

[c.] Carmustine, etoposide, cytarabine and melphalan

[d] Stem cell transplantation

[e] Hospital anxiety and depression scale

[f.] C-reactive protein

[g] Standard deviation

[h] Low Density Lipoprotein

[i] High Density Lipoprotein

[j] Volume of Oxygen

[k] Receptor Antagonist. P-values obtained by X$^2$-test for categorical variables and independent t-test or Mann-Whitney (skewed data) for continuous variables.

*One

† 4, and

‡ 2 patients missing information.

Several key enzymes of the kynurenine pathway require PLP as a cofactor and are regulated by inflammatory cytokines [37]. α-Ketoglutaric acid is also connected to glutamic acid metabolism through a PLP-dependent transamination reaction. Whereas individual vitamin B6 metabolites were not significantly different in fatigued versus non-fatigued lymphoma survivors, the PAr index was significantly higher in the CF group (mean 0.638 (95% CI 0.543–0.734)) compared to the group without CF (0.491 (0.449–0.534); p = 0.006; Fig 3). The difference was present and significant in both male and female survivors (S4 Table).

Neopterin is a catabolic product of guanosine triphosphate, a purine nucleotide and belongs to the chemical group known as pteridines. It is synthesized by human macrophages upon stimulation with the cytokine interferonγ and serves as a marker of cellular immune activation [38]. The mean (95% CI) serum concentration of neopterin in fatigued survivors was higher (27.1 (23.1–31.2) nM) than in the non-fatigued counterpart (23.1 (21.6–24.7) nM) with a borderline significance level of p = 0.07 (Fig 3). The difference was more prominent in male survivors than in females (S4 Table).

Kyn/Trp ratio, PAr index and neopterin concentrations were positively and significantly correlated with $r_s$ values of 0.57 for PAr index versus neopterin (p<0.01), 0.60 for PAr Index versus Kyn/Trp (p<0.01) and 0.80 for neopterin versus Kyn/Trp (p<0.01, S1 Fig). All three correlated with CRP levels with $r_s$ 0.28 for PAr index versus CRP (p<0.001), 0.12 for neopterin versus CRP (p = 0.06) and 0.14 for Kyn/Trp versus CRP (p = 0.03).

Since the prevalence of CF was associated with the levels of IL-6 and IL-1RA (Table 1) we analyzed the distribution of the three markers PAr index, Kyn/Trp ratio and neopterin levels according to the levels of these cytokines. All three markers were significantly higher in survivors with detectable IL-6 levels in serum compared to patients with no detectable serum IL-6 (p<0.001 for all three comparisons; Fig 4). The plasma levels of IL-1RA were significantly

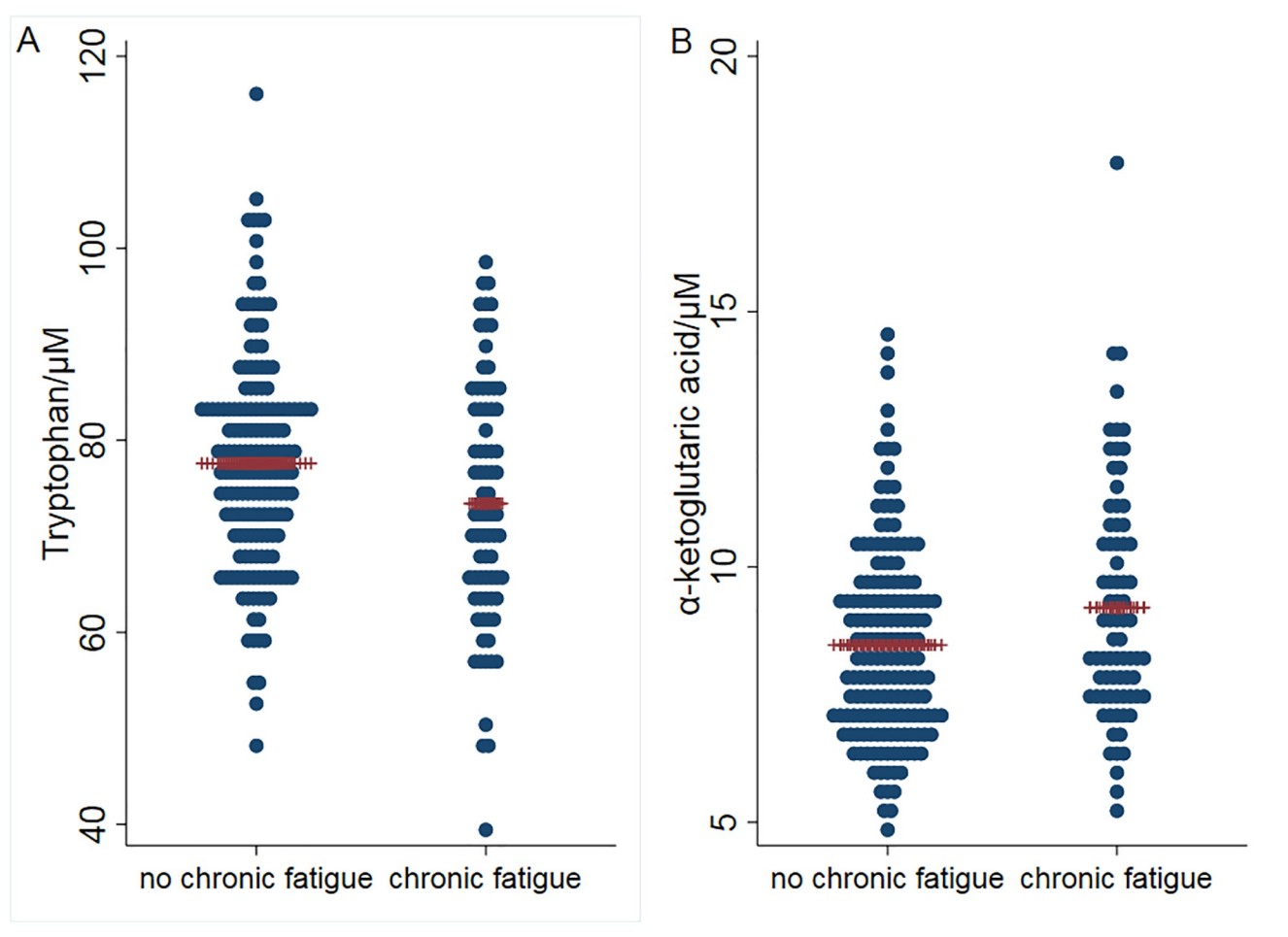

**Fig 2.** Concentrations of tryptophan (A) and α-ketoglutaric acid (B) in survivors with and without chronic fatigue. Blue dots represent individual patients; red lines represent mean values in each group. p < 0.05 for both comparisons.

correlated with PAr index ($r_s$ of 0.14; p = 0.03), Kyn/Trp ratio ($r_s$ of 0.16; p = 0.016) and neopterin ($r_s$ of 0.14; p = 0.03).

### Body mass index, nutritional status and vitamin supplements

To check for possible confounding influences of obesity, nutritional status and supplementary intake of vitamins subgroup analyses were performed. BMI and triglyceride levels correlated significantly with neither Trp levels, Kyn/Trp ratio, PAr index nor neopterin concentrations. Albumin levels, frequently used as a marker of nutritional status despite being also associated with an inflammatory response, were significantly correlated with Trp concentration ($r_s$ = 0.340; p<0.001), neopterin ($r_s$ = -0.262; p<0.001), Kyn/Trp ratio ($r_s$ = -0.350; p<0.001) and PAr index ($r_s$ = -0.352; p<0.001). Albumin was not correlated with BMI or triglycerides, but negatively correlated with CRP ($r_s$ = -0.300; p<0.001). 8 patients reported regular intake of vitamin B supplements, in most casesmixtures of different vitamins including vitamin B6. Of these 8 patients, 4 were fatigued and there were no statistically significant differences compared to those not taking vitamin B supplements neither for Trp and neopterin concentrations, Kyn/Trp ratio nor PAr index.

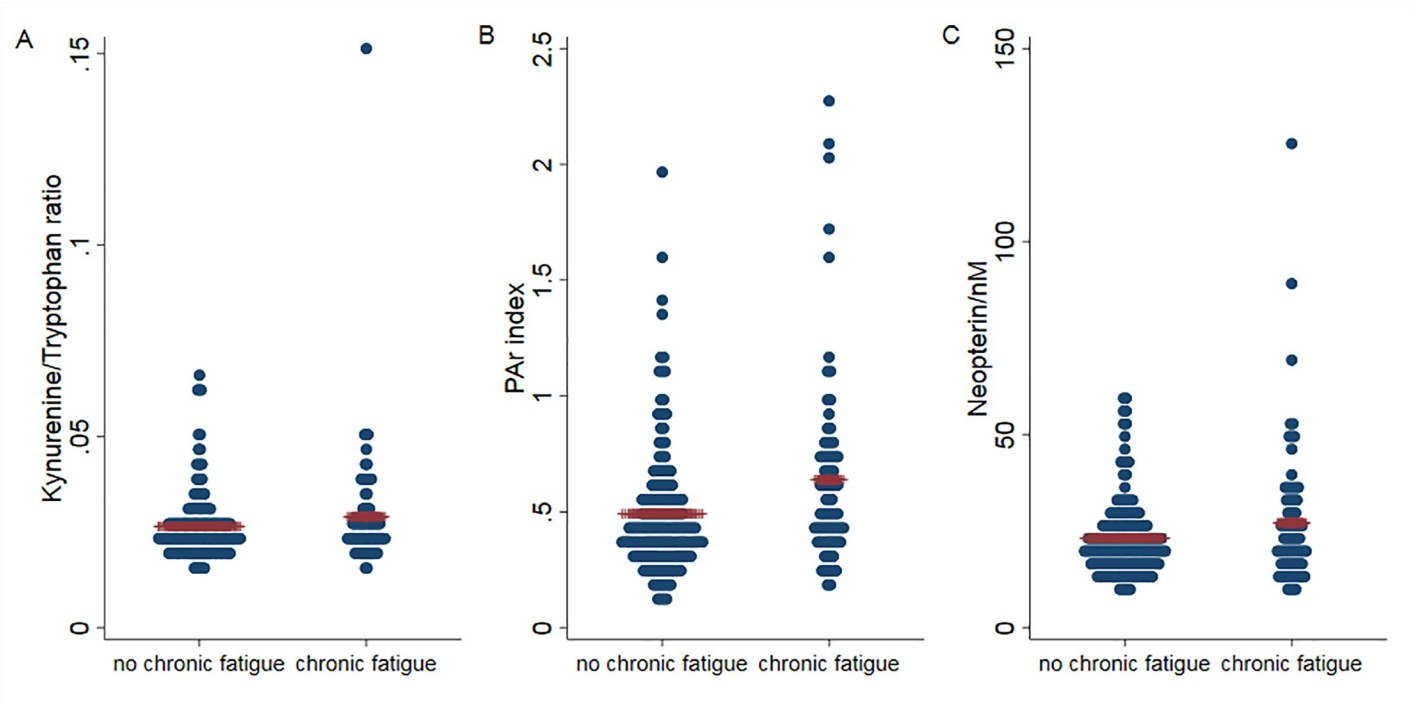

**Fig 3.** Kyn/Trp ratio (A), PAr index (B) and neopterin (C) in survivors with or without chronic fatigue. Blue dots represent individual patients; red lines represent mean values in each group. P = 0.06 for Kyn/Trp ration, p = 0.006 for PAr index and p = 0.07 for neopterin.

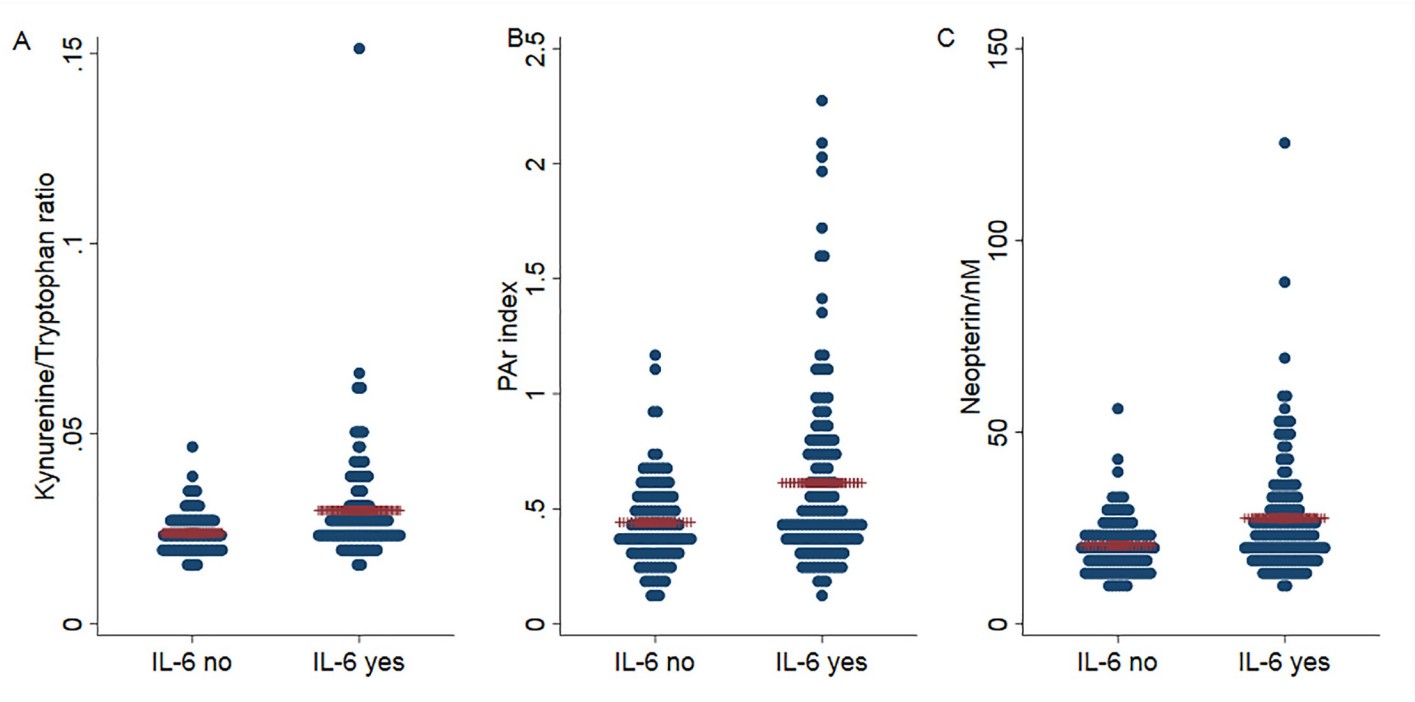

**Fig 4.** Kyn/Trp ratio (A), PAr index (B) and neopterin (C) in survivors with or without detectable IL-6 levels. Blue dots represent individual patients; red lines represent mean values in each group. p < 0.001 for all three comparisons.

## Logistic regression with chronic fatigue as dependent variable

Logistic regression with CF as dependent variable was performed as described previously [22]. Factors associated with CF with p≤ 0.10 in univariate analyses were entered in the model, but due to high correlation with each other, detectable IL-6 levels were replaced by PAr values in the model (S5 Table). In multivariable analysis, higher neuroticism score (Odds ratio (OR) = 1.50, 95% CI 1.21–1.87, p<0.001), obesity (OR = 3.11, (1.22–7.97), p = 0.02) and higher PAr values (OR = 3.62, (1.05–12.46), p = 0.04) were associated with increased risk of CF.

## Discussion

We report metabolic profiling of lymphoma survivors after HDT-ASCT with or without CF and well characterized in terms of other associated late complications, both somatic and psychological [22]. The range of analyzed metabolites was focused on amino acids and pathways related to inflammation, immune activation and vitamin B6.

The study was planned to evaluate possible changes in amino acid concentrations in survivors with CF as previously described in patients with CFS/ME [18, 19]. In CFS/ME these changes have been indicative of a defect in the tricarboxylic acid cycle, and have been postulated to play a role in the pathogenesis of this disorder. Importantly, the defects in energy metabolism could be responsible for post exertional malaise, a key clinical finding associated with fatigue in CFS/ME. No similar differences in amino acid concentrations were found when we compared lymphoma survivors after HDT-ASCT that suffer from CF to survivors without CF. The blood samples were taken under standardized fasting conditions. Since we are interested in diagnostic biomarkers of CF, potentially with relevance to pathogenic mechanisms, we deliberately chose to compare survivors with or without fatigue. As gender differences have been described in CFS/ME, we did all analyses in males and females separately. Since levels of amino acids and vitamins of the B- family may depend on nutritional status and intake of dietary supplements, relevant differences in serum concentrations were analyzed also with regard to BMI, other proposed blood markers of nutrition and against self-reported intake of vitamin supplements. The absence of similar changes of serum amino acid levels in the present study indicates different pathomechanisms in lymphoma survivors with CF and patients with CFS/ME, explaining also the clinical differences between the two conditions, i.e. female preponderance and presence of post exertional malaise, a marked aggravation of symptoms after exercise, in CFS/ME [4].

Interestingly, the difference in Trp levels observed in lymphoma survivors with or without CF (mean values 73.4 μM and 77.6 μM, respectively), were also seen when comparing patients with CFS/ME (mean concentration 73.4 μM) to healthy age matched individuals (77.8 μM) [19]. For the other amino acids tested, the lymphoma survivors resemble more closely the healthy control subjects of the latter study, irrespective of a diagnosis of CF or not [19].

Compared to their non-fatigued counterparts, survivors with CF are characterized by metabolic changes that may be associated with a low grade inflammatory status. The essential amino acid Trp is the precursor of melatonin and serotonin, and has received much attention in investigations of depression and other psychiatric disorders [39, 40]. Trp is mainly catabolized along the kynurenine pathway that produces intermediate compounds, collectively referred to as kynurenines, with a variety of effects, including neuro- and immunomodulation [37]. Metabolites of the kynurenine pathway also serve as precursors for nicotinamide adenine dinucleotide synthesis and thereby play a role in energy homeostasis. The hepatic enzyme tryptophan-2,3-dioxygenase (TDO) and the more widely expressed indolamine 2,3-dioxygenases (IDO) 1 and 2 catalyze the oxidative cleavage of Trp to Kyn, the first and rate limiting step of the pathway [41, 42]. The ratio of Kyn to Trp has been proposed as a marker of these

enzymatic reactions, but other factors, especially the activity of downstream enzymes metabolizing Kyn are also important in determining the Kyn/Trp ratio [41]. Both TDO and IDO have been implicated in a variety of disease states, including cancer and inflammation [42]. Of note, TDO activity in liver may be induced by cortisol and IDO and other enzymes of the kynurenine pathway may be activated by proinflammatory cytokines such as Interferonγ, IL-6 or TNFα [37, 43, 44]. We found that the Kyn/Trp ratio in all lymphoma survivors analyzed was significantly correlated with other markers of immune activation and inflammation, such as measurable levels of IL-6, neopterin and CRP. There was a trend towards higher Kyn/Trp ratios in survivors with CF but the difference did not reach the predefined significance level. We also found trends for different levels of two intermediates in the kynurenine pathway (HK and HAA), suggesting that the lower Trp level in serum of survivors with CF possibly is associated with altered flux through this pathway. Apparently, this pathway may also be site for gender-specific effects, as the effect on HK primarily was driven by the female patients (p = 0.04) whereas the effect on HAA primarily was associated with male patients (p = 0.07). Such effects may be explained by specific effects on the enzymes involved in this pathway, and changes in the different metabolic routes kynurenine may take. In summary, it can be speculated that low grade immune activation and inflammation cause a metabolic shift involving increased drainage of Trp from blood, through effects on the kynurenine pathway. Changes in kynurenine metabolism and elevation of the Kyn/Trp ratio are also found in chronic diseases where low grade inflammation is assumed to be essential in the pathogenesis, such as cardiovascular disease, inflammatory bowel disease, obesity and depression [45–48].

Similarly, we observed changes in the metabolism of vitamin B6 in fatigued lymphoma survivors. PLP, the active vitamer, serves as a cofactor for more than 150 enzymes including several of the enzymes involved in the kynurenine pathway. In recent years, the important function of vitamin B6 homeostasis in inflammation and immune responses in humans has attracted much interest [37]. Circulating concentrations of PLP show inverse correlations with risk and severity of a variety of diseases such as cardiovascular disease, cancer, rheumatoid arthritis and inflammatory bowel disease, all conditions in which inflammation is believed to play a key role in pathogenesis or disease progression. Blood levels of PA, the catabolite of PLP, seem positively correlated to markers of immune activation. Thus the index PAr, i.e. PA/ (PL+PLP), has been introduced as a robust marker that reflects key processes related to an individual's vitamin B6 homeostasis [49]. The PAr index is less influenced by smoking, renal function and vitamin B6 intake compared to levels of individual B6 vitamers, and is reported to discriminate efficiently patients with a high inflammatory state [49]. We found that survivors with CF had significantly higher PAr values than non-fatigued individuals, and that the PAr values where positively correlated with other markers of immune activations, such as measurable levels of IL-6, neopterin, CRP and the Kyn/Trp ratio.

Analyzing the same cohort of lymphoma survivors, we have previously reported that detectable levels of IL-6 in serum are independent determinants of CF [22]. The high correlation of IL-6 levels and both Kyn/Trp ratio, PAr index and neopterin indicate that the metabolic changes we describe are reflective of inflammation and/or immune activation in survivors with CF. PAr index was associated with the risk of CF in multivariate analysis when used independently of the other markers of inflammation or immune activation.

The association of indicators of low-grade inflammation and symptoms of CF does not allow a conclusion as to the causal direction. It could be that inflammatory processes induce fatigue in a subset of survivors, but alternatively, secondary effects of being fatigued may result in similar metabolic changes. For instance, it could be hypothesized that activation of the hypothalamus-pituitary-adrenal axis in survivors with CF may contribute to inflammation and cortisol induced TDO activation in the liver.

The changes in Trp, Kyn/Trp and PAr index in CF reported herein are novel observations and hypothesis generating, and as such, need to be validated in independent cohorts of cancer survivors. Before validation, important limitations in our observations need to be acknowledged. Our cohort consists of large number of survivors after high dose therapy for lymphoma only, all in remission at the time of analysis. The cohort is still heterogeneous in terms of lymphoma entities, additional therapy for lymphoma relapse after high dose therapy and in terms of other medical complications after treatment. The metabolic changes observed do not appear to be useful for development of diagnostic tests, neither alone nor in combinations. All differences of individual levels of amino acids, their metabolites or B6 vitamers were modest and concentrations in both fatigued and non-fatigued survivors were mostly within the normal range. There was also considerable overlap between the groups of survivors in each of the metabolites of interest, both Trp levels, Kyn/Trp ratios, individual levels of different kynurenins and the PAr index. Repeating the analyses for differences between fatigued and non-fatigued survivors after omitting possible outliers by the interquartile range rule, did however not alter the conclusion; levels of Trp, alpha-ketoglutarate and the PAr Index remained significantly associated with CF and Kyn/Trp ratios were of borderline significance only. Furthermore, we have analyzed only serum levels, and analyses of other compartments, such as intracellular levels in specific organs or cells, or other extracellular compartments such as cerebrospinal fluid, may be more informative. Due to the complex interplay of metabolic and immune pathways, more sophisticated analyses of a number of different markers together, such as network analyzes, may also be warranted [50].

## Supporting information

**S1 Table. Amino acid concentrations in lymphoma survivors.**
(DOCX)

**S2 Table. Arginine and tricarboxylic acid cycle metabolites in lymphoma survivors.**
(DOCX)

**S3 Table. Metabolites of kynurenine pathway in lymphoma survivors.**
(DOCX)

**S4 Table. Neopterin and metabolites of vitamin B6 in lymphoma survivors.**
(DOCX)

**S5 Table. Logistic regression analyses with chronic fatigue as dependent variable.**
(DOCX)

**S1 Fig.** Correlations of neopterin (A) and kynurenine/tryptophan ratio (B) with PAr index (ratio of 4-pyridoxic acid divided by sum of concentrations of pyridoxal 5'-phosphate and pyridoxal) in survivors.
(DOCX)

## Author Contributions

**Conceptualization:** Alexander Fosså, Øystein Fluge, Karl Johan Tronstad, Jon Håvard Loge, Cecilie Essholt Kiserud.

**Data curation:** Alexander Fosså, Knut Halvor Smeland, Øystein Fluge, Karl Johan Tronstad, Jon Håvard Loge, Per Magne Ueland, Cecilie Essholt Kiserud.

**Formal analysis:** Alexander Fosså.

**Funding acquisition:** Cecilie Essholt Kiserud.

**Investigation:** Alexander Fosså, Knut Halvor Smeland, Karl Johan Tronstad.

**Methodology:** Karl Johan Tronstad, Øivind Midttun, Per Magne Ueland.

**Project administration:** Alexander Fosså, Knut Halvor Smeland, Jon Håvard Loge.

**Resources:** Cecilie Essholt Kiserud.

**Software:** Knut Halvor Smeland.

**Visualization:** Knut Halvor Smeland.

**Writing – original draft:** Alexander Fosså.

**Writing – review & editing:** Alexander Fosså, Knut Halvor Smeland, Øystein Fluge, Karl Johan Tronstad, Jon Håvard Loge, Øivind Midttun, Per Magne Ueland, Cecilie Essholt Kiserud.

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
