## [Decision Letter · Decision Letter 0]

30 Oct 2019

PONE-D-19-21227

Metabolic analysis of amino acids and vitamin B6 pathways in lymphoma survivors with cancer related chronic fatigue

PLOS ONE

Dear Dr Fossa,

Thank you for submitting your manuscript to PLOS ONE. After careful consideration, we feel that it has merit but does not fully meet PLOS ONE’s publication criteria as it currently stands. Therefore, we invite you to submit a revised version of the manuscript that addresses the points raised during the review process.

We would appreciate receiving your revised manuscript by Dec 14 2019 11:59PM. To enhance the reproducibility of your results, we recommend that if applicable you deposit your laboratory protocols in protocols.io, where a protocol can be assigned its own identifier (DOI) such that it can be cited independently in the future. For instructions see: http://journals.plos.org/plosone/s/submission-guidelines#loc-laboratory-protocols

We look forward to receiving your revised manuscript.

Kind regards,

Gilles J. Guillemin

Academic Editor

PLOS ONE

Journal Requirements:

2. Please include the full name of the IRB/ethics committee that reviewed and approved this study, including the name of the affiliated institution if applicable. We additionally ask that you include your IRB/ethics committee approval number in your ethics statement

PMU is a member of the steering board of the nonprofit foundation which owns Bevital, and R&D director of Bevital, the company that carried out biochemical analyses. The authors declare no competing financial interests.

Reviewers' comments:

Reviewer's Responses to Questions

**Comments to the Author**

1. Is the manuscript technically sound, and do the data support the conclusions?

Reviewer #1: Yes

Reviewer #2: No

2. Has the statistical analysis been performed appropriately and rigorously? 

Reviewer #1: I Don't Know

Reviewer #2: No

3. Have the authors made all data underlying the findings in their manuscript fully available?

Reviewer #1: Yes

Reviewer #2: No

4. Is the manuscript presented in an intelligible fashion and written in standard English?

Reviewer #1: Yes

Reviewer #2: Yes

5. Review Comments to the Author

Reviewer #1: This is a well written research article with a clear experimental approach and well-presented results. The research area is indeed an interesting area with minimal data available.

Here are two comments/questions for the authors:

1. As IFN-y and TNF-alpha are shown to be potent inducers of the pathway, do the authors have data on these cytokines in patient’s blood? It will definitely be interesting to see if there is any correlation between these cytokines and activation of this pathway.

2. Does the author have data on the melatonin/serotonin pathway? This is so as tryptophan is the substrate for both the melatonin and kynurenine pathway.

Reviewer #2: In this paper, the authors report decreased levels of circulating tryptophan in chronically fatigued versus non fatigued cancer patients, associated with changes in metabolites of B6 and trending changes in neopterin. Despite its cross-sectional nature the study is a priori interesting as it includes many more patients than what is usually the case in clinical studies of cancer-related fatigue. Unfortunately the study does not hold to its premises because of a number of major weaknesses. Although the reported study is presented as exploratory it fails to adjust for multiplicity to account for the multiple statistical tests carried out in the same patients, which therefore gives limited credibility to the results that are reported. This is made even more problematic by the lack of any post hoc power analysis. Concerning the significance of the observed changes, there are several issues that are not considered including nutritional status and possibility of activation of TDO rather than IDO to account for the observed differences in TRP and KYN. Note that figures 1 to 3 show the existence of outliers that probably drive the differences between fatigued and non-fatigued individuals. Apparently the authors did not attempt to assess whether their results remain identical if the outliers are excluded. In discussion of the trending results the authors omit to consider the possibility of reverse causality, i.e., "stress"-related fatigue causing low grade inflammation (or HPA axis activation) which itself would be responsible for activation of the kynurenine pathway.

6. PLOS authors have the option to publish the peer review history of their article (what does this mean?). If published, this will include your full peer review and any attached files.

Reviewer #1: No

Reviewer #2: No

---

## [Author Response · Author response to Decision Letter 0]

10 Dec 2019

Response to review comments to the author

Reviewer #1: This is a well written research article with a clear experimental approach and well-presented results. The research area is indeed an interesting area with minimal data available.

Response: Thank you. We appreciate the recognition of our work.

Here are two comments/questions for the authors:

1. As IFN-y and TNF-alpha are shown to be potent inducers of the pathway, do the authors have data on these cytokines in patient’s blood? It will definitely be interesting to see if there is any correlation between these cytokines and activation of this pathway.

Response: We have analyzed TNF alfa, IL-1RA and IL-1Beta in this cohort and reported that previously (Smeland KB et al, 2019, reference 22). Neither TNF alpha nor IL-1 Beta are correlated with the presence of CF, and these cytokines are not considered in this paper. Only IL-1R alpha is correlated with fatigue. Both PAr index, Kyn/Trp ratio and neopterin are significantly correlated with the level of IL-1RA in this cohort. We have added these data in Table 1 and a separate paragraph in the result section.

2. Does the author have data on the melatonin/serotonin pathway? This is so as tryptophan is the substrate for both the melatonin and kynurenine pathway.

Response: Unfortunately, we don’t have these data. As for IFN-γ, we agree that this would be interesting.

Reviewer #2: In this paper, the authors report decreased levels of circulating tryptophan in chronically fatigued versus non fatigued cancer patients, associated with changes in metabolites of B6 and trending changes in neopterin. Despite its cross-sectional nature the study is a priori interesting as it includes many more patients than what is usually the case in clinical studies of cancer-related fatigue. Unfortunately the study does not hold to its premises because of a number of major weaknesses. 

1. Although the reported study is presented as exploratory it fails to adjust for multiplicity to account for the multiple statistical tests carried out in the same patients, which therefore gives limited credibility to the results that are reported. This is made even more problematic by the lack of any post hoc power analysis. 

Response: In deed, we view our analyses as exploratory and in need of validation it other cohorts of cancer survivors with CF. This is clearly stated in the manuscript. The design of the study was to test whether we could find similar changes in metabolites in CF as in CFS/ME patients previously reported by our group. We did not find these patterns of amino acid changes, despite a reasonably large number of patients with or without CF. As such; the study is negative and clearly stated as such in results and discussion. These differences to CFS/ME are however in themselves important as they may point to different disease mechanisms. 

The only finding that is confirmed when comparing across studies CFS/ME patients and cancer survivors with CF are the lower values for tryptophan. The analyses of the kynurenine pathway and the vitamin B6 metabolites are conducted in an effort to explain these differences, i.e. they are explanatory. As they all point in similar directions, i.e. implicate low grade inflammation in the pathogenesis of CF, we believe the data deserve to be presented at such awaiting formal validation in the future.

When it comes to post hoc power analyses, we believe that stating mean differences, standard deviations and p-values of appropriate (parametric or non-parametric tests) allow for assessment of possible type II errors, that is the risk that there are differences that would be statically significant in larger data sets. Post-hoc power analyses are in our view alternative ways to present the same information. For a broader discussion of the value of post-hoc power analyses our statistician has referred to the following web-site which we as clinician-scientists found useful to remind us of this fact.

http://daniellakens.blogspot.com/2014/12/observed-power-and-what-to-do-if-your.html

2. Concerning the significance of the observed changes, there are several issues that are not considered including nutritional status and possibility of activation of TDO rather than IDO to account for the observed differences in TRP and KYN. 

Response: We have done a number of analyses to look at nutritional status and vitamin supplements and find that these do not influence the results. This is clearly stated in the manuscript as it is. 

The origin of the proposed increased metabolism though this pathway is not known. It could be IDO or TDO, a hepatic enzyme differently regulated than IDO 1 and 2. We have added sentences in the discussion to discussion to incorporate the role of TDO in view of recent findings and clarifying reviews by experts in the field.

3. Note that figures 1 to 3 show the existence of outliers that probably drive the differences between fatigued and non-fatigued individuals. Apparently the authors did not attempt to assess whether their results remain identical if the outliers are excluded. 

Response: We do not fully agree that there are obvious outliers in most of the analyses done. We have however repeated the analyses presented for Trp, alpha-ketoglutarate, Kyn/Trp ratio, Neopterin and PAr index related to presence of absence of CF, and related to measurable IL-6 levels after excluding potential outliers by the interquartile range rule. This excludes up to 2 % of the patients from the analyses, depending on which univariate test we are looking at. The results remain stable, that is Trp is significantly reduced and both alpha ketoglutarate and PAr index both significantly increased in patients with CF compared to non-fatigued survivors. For neopterin and Kyn/Trp ration the values remain borderline with p-values of 0.06-0.12. The association with measurable IL-6 as presented in Figure 4 remains highly significant for all comparisons despite omitting the most extreme values at both ends. These data are not presented.

4. In discussion of the trending results the authors omit to consider the possibility of reverse causality, i.e., "stress"-related fatigue causing low grade inflammation (or HPA axis activation) which itself would be responsible for activation of the kynurenine pathway.

Response: We agree that this could be the case and we have added a paragraph about reverse causality in the discussion, also including a potential role of cortisol induced TDO activity.

Kind regards,

Alexander Fosså

---

## [Editor Report · Decision Letter 1]

18 Dec 2019

Metabolic analysis of amino acids and vitamin B6 pathways in lymphoma survivors with cancer related chronic fatigue

PONE-D-19-21227R1

Dear Dr. Fossa,

We are pleased to inform you that your manuscript has been judged scientifically suitable for publication and will be formally accepted for publication once it complies with all outstanding technical requirements.

With kind regards,

Gilles J. Guillemin

Academic Editor

PLOS ONE
---

## [Editor Report · Acceptance letter]

26 Dec 2019

PONE-D-19-21227R1 

Metabolic analysis of amino acids and vitamin B6 pathways in lymphoma survivors with cancer related chronic fatigue 

Dear Dr. Fosså:

I am pleased to inform you that your manuscript has been deemed suitable for publication in PLOS ONE. Congratulations! Your manuscript is now with our production department. 

With kind regards,

on behalf of

Professor Gilles J. Guillemin 

Academic Editor

PLOS ONE